# Photodynamic Effects with 5-Aminolevulinic Acid on Cytokines and Exosomes in Human Peripheral Blood Mononuclear Cells from Patients with Crohn’s Disease

**DOI:** 10.3390/ijms24054554

**Published:** 2023-02-25

**Authors:** Kristian Espeland, Andrius Kleinauskas, Petras Juzenas, Sagar Darvekar, Vlada Vasovic, Trond Warloe, Eidi Christensen, Jørgen Jahnsen, Qian Peng

**Affiliations:** 1Department of Gastroenterology, Akershus University Hospital, 1478 Lorenskog, Norway; 2Department of Pathology, Norwegian Radium Hospital, Oslo University Hospital, 0310 Oslo, Norway; 3Institute of Clinical Medicine, Faculty of Medicine, University of Oslo, 0372 Oslo, Norway; 4Department of Clinical and Molecular Medicine, Norwegian University of Science and Technology, 7030 Trondheim, Norway; 5Department of Dermatology, St. Olavs Hospital, Trondheim University Hospital, 7030 Trondheim, Norway; 6Department of Optical Science and Engineering, School of Information Science and Technology, Fudan University, Shanghai 200433, China

**Keywords:** 5-aminolevulinic acid (ALA), protoporphyrin IX (PpIX), photodynamic therapy (PDT), extracorporeal photopheresis (ECP), peripheral blood mononuclear cells (PBMCs), cytokine, exosome, flow cytometry, Crohn’s disease, autoimmune disease

## Abstract

Photodynamic therapy (PDT) using 5-aminolevulinic acid (ALA) which is the precursor of the photosensitizer protoporphyrin IX (PpIX) is an available treatment for several diseases. ALA-PDT induces the apoptosis and necrosis of target lesions. We have recently reported the effects of ALA-PDT on cytokines and exosomes of human healthy peripheral blood mononuclear cells (PBMCs). This study has investigated the ALA-PDT-mediated effects on PBMC subsets from patients with active Crohn’s disease (CD). No effects on lymphocyte survival after ALA-PDT were observed, although the survival of CD3^−^/CD19^+^ B-cells seemed slightly reduced in some samples. Interestingly, ALA-PDT clearly killed monocytes. The subcellular levels of cytokines and exosomes associated with inflammation were widely downregulated, which is consistent with our previous findings in PBMCs from healthy human subjects. These results suggest that ALA-PDT may be a potential treatment candidate for CD and other immune-mediated diseases.

## 1. Introduction

Photodynamic therapy (PDT) is an established treatment option for some pre-malignant and malignant diseases. The combination of a photosensitizing agent that localizes in a lesion and light together with oxygen can induce both photochemical and photobiological reactions leading to irreversible photodamage to the lesion through apoptosis and necrosis of diseased cells [1,2,3]. 

PDT using the endogenous protoporphyrin IX (PpIX) is generally well tolerated and has fewer side-effects than chemically synthesized photosensitizers. 5-Aminolevulinic acid (ALA), an amino acid, is a naturally occurring precursor to PpIX. The supply of exogenous ALA will lead to the accumulation of PpIX in proliferating cells [4,5,6,7]. This accumulation can be used for photodiagnosis (PD) and PDT [8,9,10]. Systemic use of ALA is well documented in fluorescence-guided surgical resection of glioma [11,12,13]. ALA or its derivative hexaminolevulinate is also used for PD of bladder cancer with fluorescent cystoscopy [14,15]. In addition, topical administration of ALA or its methyl aminolevulinate derivative is approved for the PDT of skin malignancies [16,17].

Cytokines are a family of peptides (with both pro- and anti-inflammatory properties). They are largely produced by immune cells to signal and modulate immune functions. Exosomes, released by all types of cells, are membrane-derived, nano-sized vesicles. They carry cargoes including lipids, proteins, nucleic acids, etc., to play a diverse role in the intercellular communications for disease pathogenesis, immune regulation, and treatment responses [18,19,20]. They may also serve as diagnostic [21] and prognostic [22] biomarkers for cancer. 

Crohn’s disease (CD) is a chronic, immune-mediated, inflammatory disease that can involve any part of the gastrointestinal tract [23,24]. The incidence is increasing in both developed and developing countries [25,26]. Mostly young adults are affected, and the disease has a significant impact on the patients in terms of both quality of life and morbidity. The exact pathogenesis of CD is not fully understood yet, although factors of genetics, the environment, gut microbiota, and immune dysfunction are involved [27]. As intestinal epithelial cells are exposed to bacteria and food antigens, an immune response is needed against the pathogens. If such a response is too strong and prolonged it causes inflammatory diseases. The immune system of the body thus uses small various extracellular components including exosomes [28] to balance between immune tolerance towards normal microbes and eliciting an immune response against pathogens. Since many pro-inflammatory cytokines and exosomes are produced from immune cells in CD, current treatments of CD with different drugs focus on inhibiting the immune response [29,30]. Vedolizumab, for instance, binds selectively to activated lymphocytes in the gastrointestinal tract [31,32]. Although the symptoms in many CD patients have been improved with biological therapy, refractory patients [33] and the side effects of such treatments confirm the need for other treatment of this condition [30]. 

Extracorporeal photopheresis (ECP) with 8-methoxypsoralen (8-MOP) has been employed to treat CD [34,35,36,37]. Recently, we have also tried ALA-based ECP for the T-cell-mediated chronic graft versus host disease (cGvHD) with promising results [38,39], and our clinical trial in ALA-ECP of CD is underway. In the present study, we used peripheral blood mononuclear cells (PBMCs) from patients with untreated active CD as a model to in vitro study ALA-based photodynamic effects on the changes in cytokines and exosomes of PBMCs as biomarkers for treatment responses to better understand the possible mechanism involved in ALA-ECP of CD.

## 2. Results and Discussion

### 2.1. Sample Collection and Clinical Patient Information

Samples were taken from patients with active CD based on the clinical activity index and/or objective biomarkers of inflammation including fecal calprotectin and CRP. Additionally, the simple endoscopic score for CD (SES-CD) was used for endoscopic assessment (Figure 1, Table 1). The colonoscopy and scoring were performed in routine clinical practice and were not completed in relation to the study. The yield of isolation of PBMCs from the blood was within the range as expected [40].

### 2.2. Cell Survival after ALA-PDT

The cell survivals of lymphocytes, B-cells, and monocytes from the samples of eight different CD patients were evaluated at 48 h after PDT with ALA plus light exposure. Figure 2A shows no apparent killing effects of ALA-PDT on lymphocytes. Our pilot studies have shown that most of such lymphocytes were T-cells and there were few activated T-cells in the PBMCs of the CD patients. Since the resting T-cells are not proliferative and produce much less PpIX from ALA than activated T-cells, the photodynamic killing effect on the T-cells is minimal. In the case of B-cells, ALA alone or illumination alone did not kill the B-cells, but ALA plus light seemed to slightly reduce the B-cell survivals in some samples (Figure 2B). Figure 2C demonstrates clear killing effects of ALA-PDT on monocytes in all of the samples.

### 2.3. Effect of ALA (Alone) on Cytokines

Overall, there is a downregulation of cytokines after ALA alone (Figure 3 and Figure 4) as compared to the untreated control samples with no ALA or light. Although there are huge individual variations of the changes in the amounts of cytokines after ALA alone (Figure 3), seven selected cytokines with the lowest variabilities are presented in Figure 4. All of the seven cytokines were averagely downregulated, but some were upregulated in the samples (see max. values). Interleukin-1 receptor antagonist (IL-1ra) naturally counteracts the proinflammatory effects of IL-1. It binds to the IL-1 receptor in the acute phase of infection or inflammation as a mediator of innate immunity [41]. Vascular endothelial growth factor (VEGF) could be associated with colorectal alterations in CD [42], as it induces angiogenesis [43]. Yang and associates found upregulation of the VEGF-C/VEGFR3 pathway in the skin following ALA-PDT [44]. Interleukin-5 (IL-5), interleukin-8 (IL-8) [45], and interleukin-9 (IL-9) [46] are all involved in inflammation. As cytokines were mainly downregulated in our study, it seems ALA alone could have an anti-inflammatory effect, as found by others [47]. Eotaxin may play a role in a number of diseases including inflammatory bowel diseases such as CD [48]. Regulated upon activation normal T-cell expressed and presumably secreted (RANTES), also known as chemokine ligand 5 (CCL5), might be interesting in the treatment of inflammatory bowel disease (IBD) [49].

### 2.4. Effect of ALA and Red Light on Cytokines

After ALA-PDT a large number of cytokines were upregulated (Figure 5); although, on average, the majority of selected cytokines with the lowest variabilities were downregulated as compared to those treated with ALA alone (Figure 6). Interestingly, although ALA alone reduced the level of the anti-inflammatory cytokine, IL-1ra (Figure 5), ALA-PDT apparently increased the production of IL-1ra (Figure 6), suggesting an important role in inhibiting inflammation. Interferon gamma (IFN-γ) can potentiate pro-inflammatory signaling via immune cells, while IL-8 and macrophage inflammatory protein-1 beta (MIP1-β) are known as chemotactic factors for granulocytes and other immune cells, respectively [50]. The downregulation of these cytokines by ALA-PDT indicates anti-inflammatory effects. Several cytokines or downstream of the cytokines associated with IBD [51] were noted in this study. Monocyte chemoattractant protein 1 (MCP-1) is involved in immune-mediated diseases such as CD. Gaiani et al. found that MCP-1 together with other markers was elevated in non-responders to infliximab [52], which is a biological drug for the treatment of CD. Moreover, another biological agent ustekinumab seems to downregulate MCP-1 [53], consistent with the effect of ALA-PDT.

### 2.5. Effect of ALA (Alone) on Exosomes

The release of exosomes from cells in different conditions is a new and interesting field. Exosomes can act as drug targets and biomarkers and also function as drug delivery agents. They are available in easily accessible body fluids [54] and may play a role in the treatment of CD [55]. Technically, however, isolation of exosomes can be challenging. In this study we used a bead-based technique [56], the same as described in our published report [57].

Exosomes were largely produced after the treatment with ALA alone (Figure 7). Those with the least variabilities are the exosomes of CD24, CD29, CD40, CD31, HLA-ABC, SSEA4, and CD81 (Figure 8). Although the levels of CD146 and CD11c exosomes were averagely reduced (Figure 8), these two exosomes were actually increased in six out of eight samples (Figure 7). CD24 is mainly for B-cells and CD40 is mainly for T-cells and macrophages [58] while CD81 is for both T- and B-cells. HLA-ABC is for all nucleated cells and stage-specific embryonic antigen-4 (SSEA-4) for mesenchymal stem cells. The increases in those exosomes by ALA alone might have functional implications. CD29-expressing cells in cooperation with T-cells is a part of the inflammatory cascade in ulcerative colitis (UC) [59,60]. CD31 exosomes released from platelets, macrophages, and lymphocytes may be of importance in inflammation. The upregulation of this exosome with ALA alone followed by downregulation after ALA-PDT may be interesting for further exploration [61]. The CD146 exosome is upregulated in inflamed samples from CD patients [62]. The release of CD146 exosomes may be due to the patients with the active disease as a part of the inflammatory cascade. Lastly, CD11c cells may play an important role in the immune response in the intestines in CD [63] and are related to the severity of such inflammation.

### 2.6. Effect of ALA and Red Light on Exosomes

ALA-PDT markedly decreased most exosomes with different markers except those taken from sample 3 (Figure 9). This may be due to a direct PDT effect on exosome production from cells or the destruction of cells. On average, all of the selected exosomes with fewer variabilities were downregulated (Figure 10). CD19 and CD20 are B-cell antigens and the decreases in these types of exosomes may be consistent with the slightly reduced survivals of B-cells in some samples after ALA-PDT (Figure 2B).

CD86 is expressed in dendritic cells while CD69 and CD44 are involved in the activation of lymphocytes and natural killer cells. CD326 and SSEA-4 exosomes are biomarkers for organ transplantation rejection [64,65]. Downregulation of these two exosomes following ALA-PDT may suggest a beneficial effect and is in agreement with the results of our previous report [57]. CD105 or endoglin (ENG) found in endothelial cells is associated with angiogenesis [66] whereas CD142, CD41b, and CD42a participate in blood coagulation cascades. CD62p or p-selectin is involved in cell adhesion in the initial phase of inflammation [67,68]. Interestingly, CD9 is often used as a marker for exosomes [69,70] together with CD63 and CD81 [55]. Generally, the cells of CD19, CD20, CD24, CD29, CD31, CD40, CD44, CD62p, CD69, CD81, and CD86 are involved in the immune defense system [69,70]. Downregulation of these exosomes by ALA-PDT may inhibit the immune-mediated reactions that occur in CD patients. Since CD is an immune-mediated, inflammatory disease [23,24,25,29], it would be of interest to investigate the effects of ALA-PDT on the PBMCs of patients with other immune-mediated diseases in the future.

## 3. Materials and Methods

### 3.1. Patient Population and Inclusion of Samples

After informed consent, 12 mL of full blood from each of the 30 total patients with untreated (no biological drugs or corticosteroids) CD were collected, and plasma and PBMCs isolated as described in Section 3.3. Of these patients, three samples were excluded due to them not representing active CD (CRP < 5 mg/L and/or fecal calprotectin < 250 µg/mg) and two were excluded because of mistakenly mixed samples. The cell viabilities of the samples from the remaining 25 patients were screened. A minimal amount of live cells (2 × 10^6^) per sample was set in order to meet the requirement of the experiments. Eight samples from eight different patients were eligible for this study (Figure 1) with clinical information presented in Table 1. The remaining 17 samples from 17 patients had less than 2 × 10^6^ live cells per sample and were thus disqualified in this study. 

### 3.2. Chemicals

5-Aminolevulinic acid (ALA) was obtained from Sigma Aldrich (St. Louis, MO, USA). A fresh stock solution of ALA was prepared in phosphate buffered saline (PBS) (VWR Life Science, Solon, OH, USA) to a concentration of 1 M and kept at 4 °C. This was further diluted to reach a concentration of 3 mM for each experiment. All of the chemicals used were of the highest purity commercially available.

### 3.3. Isolation and Culture of PBMCs

Full blood samples from 30 patients with untreated active CD (Regional Committee for Medical Research Ethics, REK midt 2018/631) were obtained from the Department of Gastroenterology, Akershus University Hospital. The isolation of PBMCs from the full blood was completed using a Lymphoprep density gradient solution (Axis-Shield, Oslo, Norway) in SepMate^TM^ 50 mL tubes (Stemcell Technologies, Cambridge, UK). Fifteen mL of Lymphoprep was pipetted to one SepMate^TM^ tube. The buffy coats were diluted with an equal volume of RPMI-1640 growth medium (Gibco, Grand Island, NY, USA) containing 2% fetal bovine serum (FBS) (Biological Industries Israel Beit-Haemek Ltd., Kibbutz Beit-Haemek, Israel) and layered carefully above the density gradient solution. The tubes were centrifugated at 1200× *g* for 10 min at room temperature with the brake on. The plasma from the top layer (2–4 mL) of enriched mononuclear cells was collected and stored at −80 °C. The top layer was then poured off for maximally 2 s to a new 50 mL Corning Falcon^TM^ tube (Corning, NY, USA) and washed twice with 40 mL growth medium with 2% FBS, firstly at 300× *g* for 15 min and secondly at 300× *g* for 10 min. The isolated PBMCs were then counted with a Countess II Cell Counter (Thermofisher, Waltham, MA, USA) before being gradually frozen using a CoolCell^TM^ (Corning Incorporated, Corning, NY, USA) and stored at −80 °C before use.

The PBMCs were defrosted with the standard method and incubated in RPMI-1640 growth medium with 10% exosome-depleted FBS, 100 units/mL penicillin, 10 µg/mL streptomycin, and 2 mM L-glutamine at 37 °C in a humidified atmosphere with 5% CO_2_ (Nuaire Model NU-5810E incubator, Plymouth, MN, USA) for 24 h before the experiments were performed.

### 3.4. Light Source

A light emitting diode (LED) lamp (Aktilite^®^ CL128, Photocure ASA, Oslo, Norway) was used for the PDT experiments. The lamp peak emission wavelength was 630 nm with a fluence rate of 100 mW/cm^2^ for 30 min giving a total light dose of 180 J/cm^2^.

### 3.5. PDT Treatment of PBMCs

Two million viable PBMC cells from each patient were equally divided into four samples (5 × 10^5^ cells per sample), i.e., control, ALA alone, light ALA alone, and ALA plus light. The cells were seeded into two 24-well plates (Nunc, Thermo Fisher Scientific, Roskilde, Denmark). The two plates had a similar design with appropriate wells containing mock cells (no ALA) and treated cells (3 mM ALA). To mimic a clinical ex vivo photopheresis procedure, the plates were covered with aluminum foil and incubated at room temperature (22 °C). One plate was not illuminated as a control. Four hours after ALA incubation, the other plate was irradiated with a 630 nm LED lamp at room temperature. After light exposure, both plates were further incubated for another 48 h in the dark at 37 °C in a humidified atmosphere with 5% CO_2_ with the same medium in each well to maintain the cytokines and exosomes. Survivals of the PBMC subsets were measured with flow cytometry as described in Section 3.6.

### 3.6. Flow Cytometry Analysis

At 48 h after treatment, the cell suspensions from both plates were collected into appropriately marked 1.5 mL tubes (Eppendorf AG, Hamburg, Germany) and centrifuged at 3000 rpm for 5 min. The supernatants were carefully collected without disturbing the cell pellets and kept at −80 °C before isolation, labelling, and measurement of cytokines and exosomes as described in Section 3.7. The cell pellets were immediately used for the phenotyping experiment. The B-cells were defined as a CD3^−^/19^+^ population. A cocktail of a CD3-APC antibody (Invitrogen, Carlsbad, CA, USA), CD19-PE antibody (ImmunoTools GmbH, Friesoythe, Germany) and fixable viability dye eFluor 450 (Invitrogen) was prepared in PBS with 2% FBS. The dilutions were 1:500 for both antibodies and the viability dye. Fifty μL of the antibody cocktail was added to each cell pellet and thoroughly mixed. The cells were incubated for 1 h at room temperature with occasional vortexing. The samples were then washed once with 1 mL PBS per sample and resuspended in 100 μL PBS with 2% FBS for flow cytometry assay. The survival of B-cells was determined with the antibody cocktail while the viabilities of the lymphocytes and monocytes were assessed using only the viability dye channel after being gated out from PBMCs in the light scatter plots. The measurements were performed using a Cytoflex S cytometer (Beckman Coulter Life Sciences, Indianapolis, IN, USA) with Cytexpert software (Version 2.1, Beckman Coulter). The data analyses were performed using FlowJo software (Version 10.5.2, Treestar, Ashland, OR, USA).

The data for the percentages of lymphocytes and monocytes (Figure 2A,C) are presented as means of three different measurements performed on three different days. The data for live cells are presented as means of two different measurements performed on two different days. The error bars for each sample show the standard deviations of these measurements. The percentages and live cells of B-lymphocytes (Figure 2B) were determined only from a single experiment and the data for each sample are shown without error bars. The gating strategy of flow cytometry for the measurements is also included as Appendix A.

### 3.7. Isolation, Labelling, and Measurement of Cytokines and Exosomes

The isolation and labelling of cytokines and exosomes were based on the Exosome Isolation Kit CD63 (Magnetic Exosome Isolation Beads) and MACSPlex Exosome Kit (MACSPlex Exosome Capture Beads, MACSPlex Exosome Detection Reagent CD63, and 3 Buffer Solutions) (both kits from MACS Miltenyi Biotech GmbH, Bergisch Gladbach, Germany), as described in our previous report [57]. On day 1, after thawing, 900 µL of a clean supernatant sample (containing both cytokines and exosomes) was transferred to an Eppendorf tube (Eppendorf AG, Hamburg, Germany). Fifty µL of Magnetic Exosome Isolation Beads were added to the sample and the mixture was put on a shaker (Vortex T Genie-2, Scientific Industries, Bohemia, NY, USA) at a vortex speed of 3 for 8 h at room temperature and stored overnight in a refrigerator at 4 °C before magnetic isolation of the exosomes.

On day 2, a proposed µ-column (same manufacturer) was placed in a magnetic µMACS separator that was attached to a MACS multistand (same manufacturer). The µ-column was added with 100 µL of the equilibration buffer (same manufacturer) followed by washing with 100 µL of the Isolation Buffer for three times. The magnetically isolated sample from day 1 was then added to go through the µ-column for cytokine collection first (with no need for the magnetic bead’s help). Subsequently, the µ-column was washed four times using 200 µL of the isolation buffer. After washing, the µ-column was moved away from the magnetic separator and immediately flushed into a clean Eppendorf tube with 100 µL of the isolation buffer by using a plunger. For labelling of the exosomes, 15 µL of the MACSPlex Exosome Capture Beads was added to the same sample and mixed carefully. A control sample without exosomes was also included using the same capture beads. The sample was then put on the shaker for 1 h and stored overnight in a refrigerator at 4 °C.

On day 3, the sample was put on the shaker at vortex speed 3 for 5 h. One mL of the MACSPlex Buffer was added to the sample and centrifuged at 12,000 rpm for 10 min. The supernatant was removed and 5 µL of the MACSPlex Exosome Detection Reagent CD63 (for the detection of surface markers of exosomes) was added before 3 h incubation on the shaker. The sample was then washed twice by adding 1 mL of the MACSPlex Buffer, centrifuged at 12,000 rpm for 10 min, and the supernatant was removed again.

Cytokines were labelled using a Bio-plex Pro human Cytokine Grp 1 panel 27-Plex kit (Bio-Rad Laboratories, Inc., Hercules, CA, USA). This was carried out by firstly mixing 50 µL of a sample with 50 µL of a bead solution in an Eppendorf tube. The bead solution was made by dilution of 2 µL beads in 50 µL assay buffer. After mixing, this was incubated for 30 min and washed by adding 1 mL of wash buffer (diluted 10 times in distilled water), centrifuged at 10,000 rpm for 5 min, and the was supernatant removed. Then, 20 µL of detection antibody solution (2 µL detection antibody in 50 µL detection antibody diluent) was added to the sample, incubated for 30 min, and washed as mentioned above before the supernatant was aspirated. Twenty µL of streptavidin—PE solution (1 µL streptavidin-PE in 100 µL detection antibody diluent) was then added and incubated for 10 min. The same wash step was repeated, and the supernatant was removed. Finally, 50 µL of the assay buffer was added per sample.

The cytokine and exosome samples were measured with a Cytoflex S cytometer as described in Section 3.6.

### 3.8. Statistical Analyses and Criteria of Data Selection

A total of 27 cytokines and 35 exosomes in each sample of eight different CD patients were measured in this study. Data were analyzed as described in the legends of the figures. Since there are considerable individual variations in the amounts of cytokines and exosomes after treatment with ALA alone or ALA plus light, we used the percentage of standard deviation of the average and set the criteria of data selection to include all those data with acceptable variabilities. They include 200% for the treatment with ALA alone and 200% with ALA plus light in the case of cytokines and 300% for the treatment with ALA alone and 100% with ALA plus light in the case of exosomes.

## 4. Conclusions

This study demonstrates a wide anti-inflammatory response at the subcellular levels of cytokines and exosomes following ALA-PDT in the PBMCs of eight different patients with active CD. The treatment with ALA alone also had effects on some cytokines and exosomes as compared to the untreated control CD samples. At the cellular level, the survivals of monocytes were clearly reduced in all of the samples. This study has a limited number of patient samples. Therefore, further studies are warranted to confirm such ALA-mediated photodynamic effects in CD and other auto-immune disorders.

## Figures and Tables

**Figure 1 ijms-24-04554-f001:**
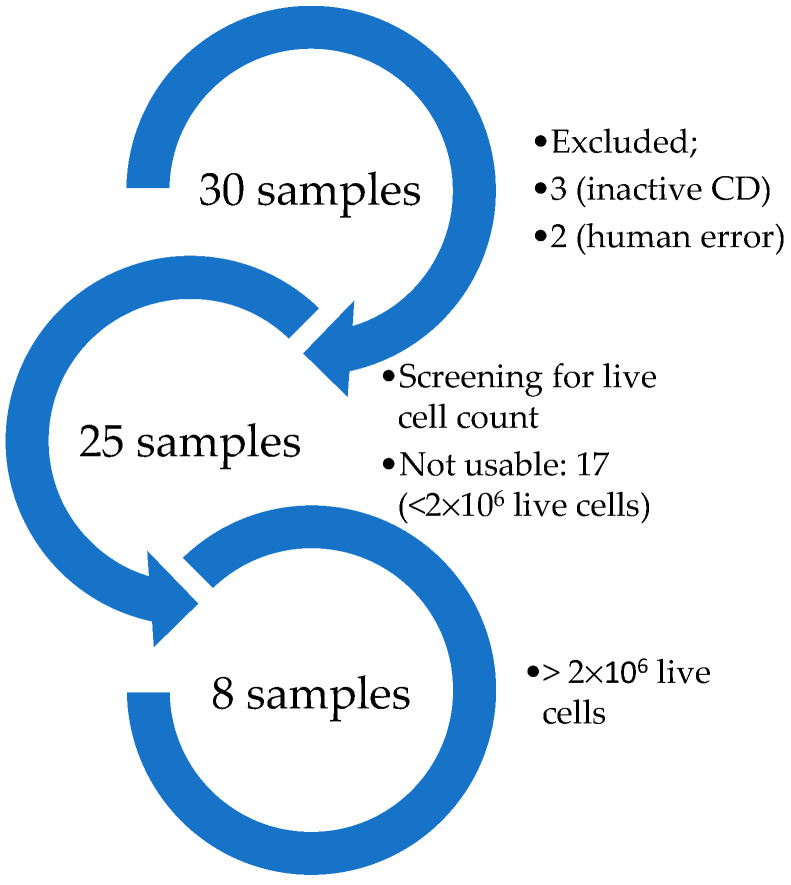
Setup for sample collection.

**Figure 2 ijms-24-04554-f002:**
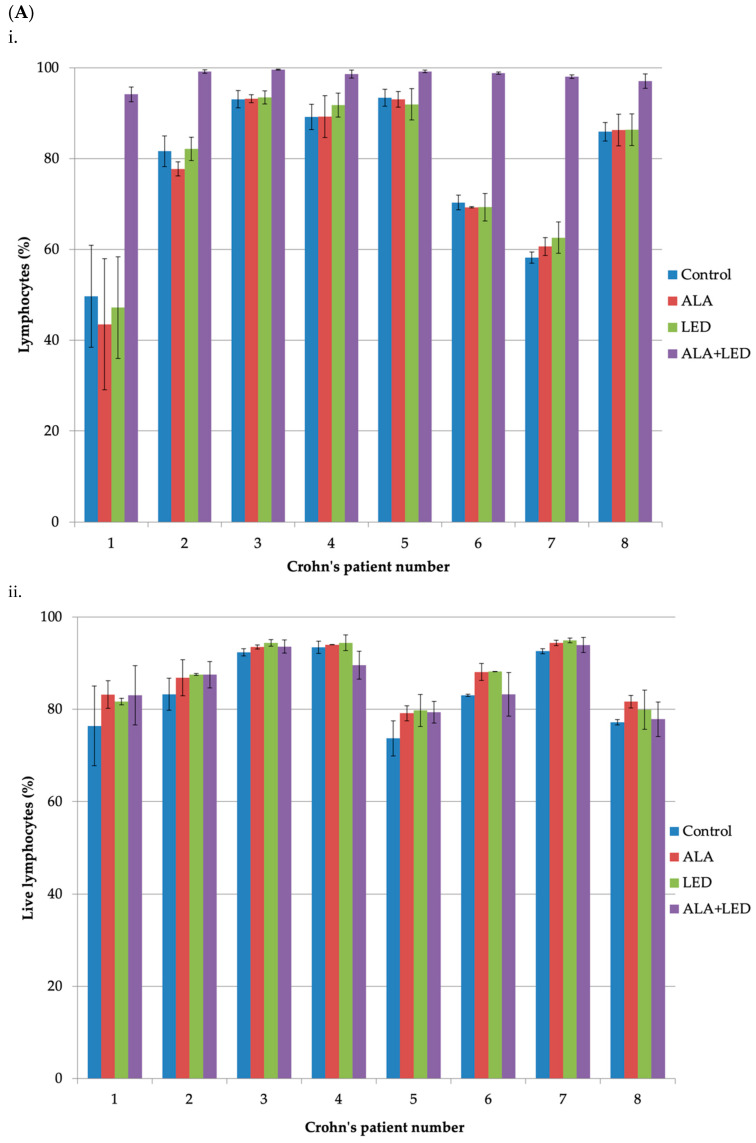
Cell survivals of PBMC subsets of CD patients after ALA-PDT in vitro. (**A**) Lymphocytes; (i) percentages of lymphocytes of PBMCs, (ii) fractions of live lymphocytes in respective patient samples in (i); (**B**) B-cells; (i) percentages of B-cells of PBMCs, (ii) fractions of live B-cells in respective patient samples in (i); and (**C**) monocytes; (i) percentages of monocytes of PBMCs, (ii) fractions of live monocytes in respective patient samples in (i). The ALA-PDT protocol is described in Section 3.5.

**Figure 3 ijms-24-04554-f003:**
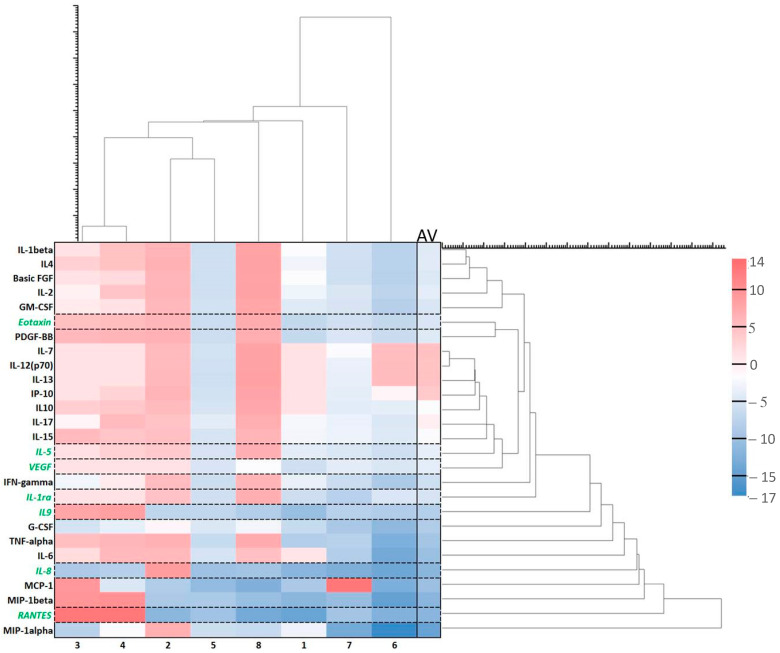
Effects of ALA alone on cytokines. Cytokine heatmap with log values. Log = sgn (ALA_i_ − Ctrl_i_) * log_2_(|ALA_i_ − Ctrl_i_|) where ALA_i_ is the cytokine fluorescence intensity in the group with ALA alone and Ctrl_i_ is the cytokine fluorescence intensity in the control group with no ALA or light. The pink colors represent increased cytokines while the blue ones show decreased cytokines after treatment with ALA alone. The heatmap is vertically clustered by log Euclidian distances and also horizontally clustered by various patient samples. The selected cytokines with the lowest variabilities are shown with the green color. AV is an average of the log values from different samples of eight patients.

**Figure 4 ijms-24-04554-f004:**
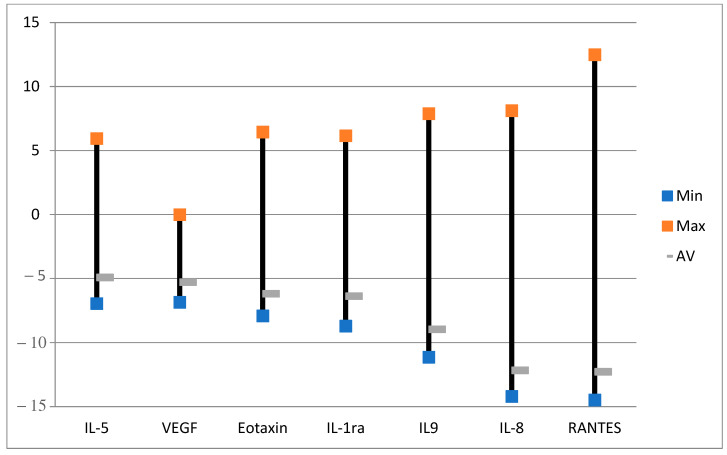
Effects of ALA alone on selected cytokines with the lowest variabilities. The data are presented as minimal, maximal, and average log values. Log = sgn (ALA_i_ − Ctrl_i_) * log_2_(|ALA_i_ − Ctrl_i_|), where ALA_i_ is the cytokine fluorescence intensity in the group with ALA alone and Ctrl_i_ is the cytokine fluorescence intensity in the control group with no ALA or light.

**Figure 5 ijms-24-04554-f005:**
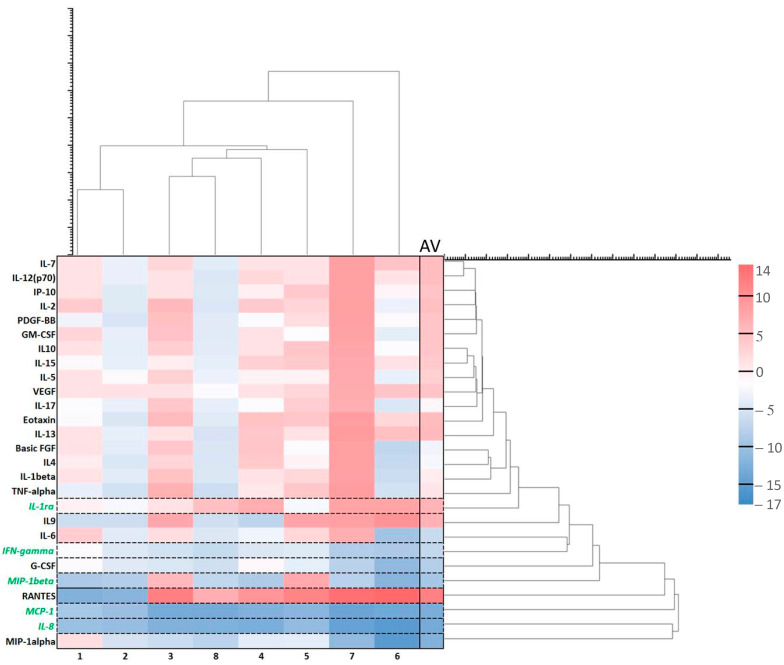
Effects of ALA and red light on cytokines. Cytokine heatmap with log values. Log = sgn(PDT_i_ − ALA_i_) * log_2_(|PDT_i_ − ALA_i_|), where PDT_i_ is the cytokine fluorescence intensity in the group with ALA plus light and ALA_i_ is the cytokine fluorescence intensity in the group with ALA alone. The pink colors represent increased cytokines while the blue ones show decreased cytokines after treatment with ALA plus light. The heatmap is vertically clustered by log Euclidian distances and also horizontally clustered by various patient samples. The selected cytokines with the lowest variabilities are shown with the green color. AV is an average of the log values from different samples of eight patients.

**Figure 6 ijms-24-04554-f006:**
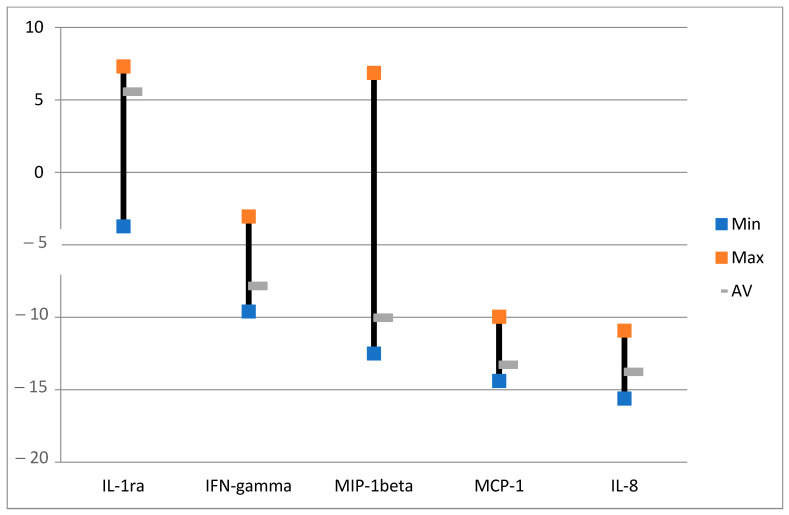
Effects of ALA and red light on selected cytokines with the lowest variabilities. The data are presented as minimal, maximal, and average log values. Log = sgn (PDT_i_ − ALA_i_) * log_2_(|PDT_i_ − ALA_i_|), where PDT_i_ is the cytokine fluorescence intensity in the PDT group with ALA plus light and ALA_i_ is the cytokine fluorescence intensity in the control group with ALA alone.

**Figure 7 ijms-24-04554-f007:**
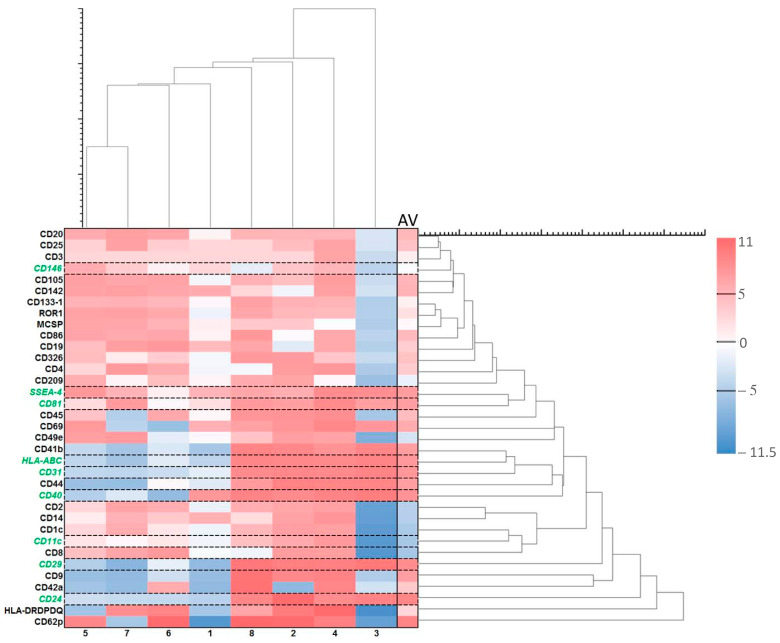
Effects of ALA alone on exosomes. Exosome heatmap with log values. Log = sgn(ALA_i_ − Ctrl_i_) * log_2_(|ALA_i_ − Ctrl_i_|), where ALA_i_ is the exosome fluorescence intensity in the group with ALA alone and Ctrl_i_ is the exosome fluorescence intensity in the control group with no ALA or light. The pink colors represent increased exosomes while the blue ones show decreased exosomes after treatment with ALA alone. The heatmap is vertically clustered by log Euclidian distances and also horizontally clustered by various patient samples. The selected exosomes with the lowest variabilities are shown with the green color. AV is an average of the log values from different samples of eight patients.

**Figure 8 ijms-24-04554-f008:**
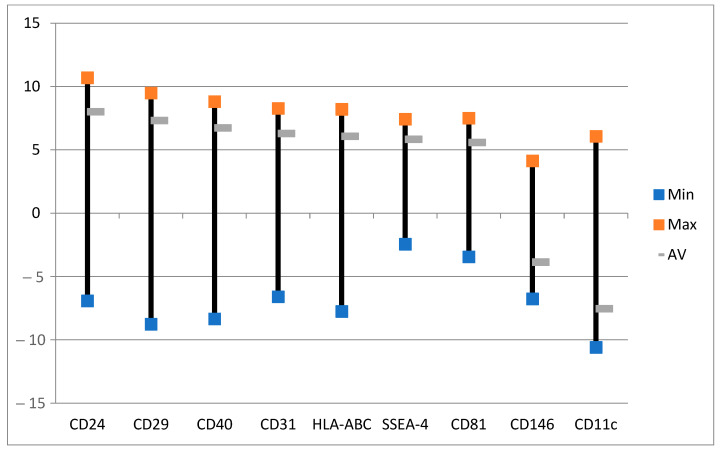
Effects of ALA alone on selected exosomes with the lowest variabilities. The data are presented as minimal, maximal, and average log values. Log = sgn (ALA_i_ − Ctrl_i_) * log_2_(|ALA_i_ − Ctrl_i_|), where ALA_i_ is the exosome fluorescence intensity in the group with ALA alone and Ctrl_i_ is the exosome fluorescence intensity in the control group with no ALA or light.

**Figure 9 ijms-24-04554-f009:**
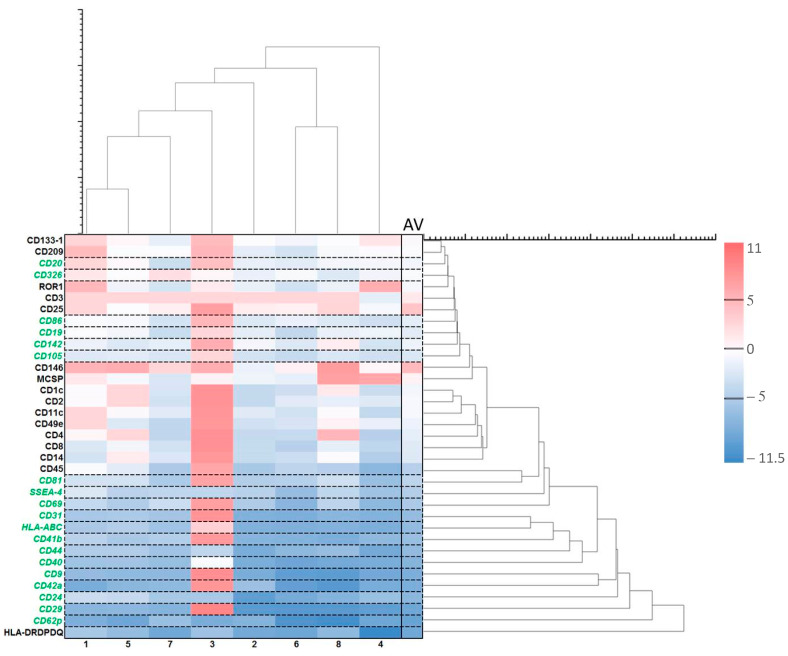
Effects of ALA and red light on exosomes. Exosome heatmap with log values. Log = sgn(PDT_i_ − ALA_i_) * log_2_(|PDT_i_ − ALA_i_|), where PDT_i_ is the exosome fluorescence intensity in the group with ALA plus light and ALA_i_ is the exosome fluorescence intensity in the group with ALA alone. The pink colors represent increased exosomes while the blue ones show decreased exosomes after treatment with ALA plus light. The heatmap is vertically clustered by log Euclidian distances and also horizontally clustered by various patient samples. The selected exosomes with the lowest variabilities are shown with the green color. AV is an average of the log values from different samples of eight patients.

**Figure 10 ijms-24-04554-f010:**
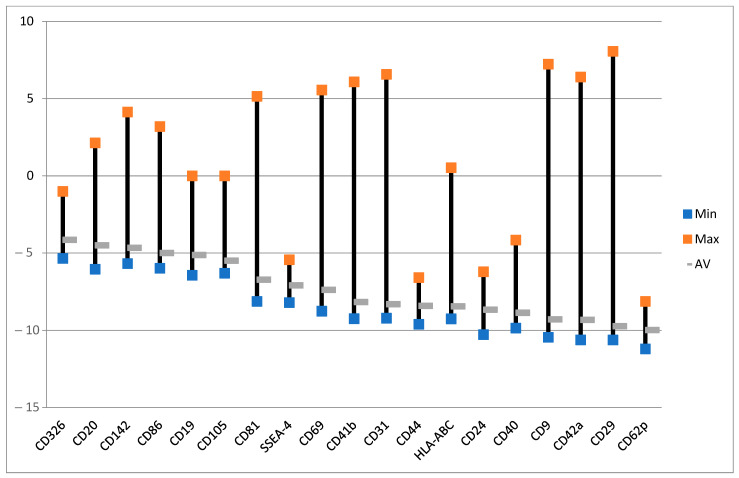
Effects of ALA and red light on the selected exosomes. Exosomes with the lowest variabilities. The data are presented as minimal, maximal, and average log values. Log = sgn (PDT_i_ − ALA_i_) * log_2_(|PDT_i_ − ALA_i_|), where PDT_i_ is the exosome fluorescence intensity in the PDT group with ALA plus light and ALA_i_ is the exosome fluorescence intensity in the control group with ALA alone.

**Table 1 ijms-24-04554-t001:** Clinical information.

Item	Mean (Range)	Reference Values
Sex	5 M, 3 F	-
Age (yrs.)	51 (30–77)	-
Fecal calprotectin (µg/mg)	1460 (542–2760)	<250
CRP (mg/L)	50 (2–130)	<5
Simple Endoscopic Score for CD (SES-CD)	10 (6–15)	<3

## Data Availability

The data presented in this study are available in the Appendix A and in Section 2 of the article.

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
