# Peer review of "Photodynamic Effects with 5-Aminolevulinic Acid on Cytokines and Exosomes in Human Peripheral Blood Mononuclear Cells from Patients with Crohn’s Disease"

_ijms, 2023, doi:10.3390/ijms24054554_

Round 1
Reviewer 1 Report
The authors present a study of the anti-inflammatory response of ALA-induced PDT in vitro. This treatment is proposed as a possible treatment avenue for autoimmune diseases such as Crohn’s disease. The authors isolated peripheral blood mononuclear cells (PBMCs) from clinical patients. They present the effect of ALA and ALA-PDT on a multitude of cytokine biomarkers and exosomes relevant to inflammation. The presented work is well executed and will be of high interest to researchers in the area of immune diseases. However, there are minor issues to address which I have presented below:
1. The authors are studying the effect of ALA and PDT on exosomes, but the significance of exosomes is not presented in the introduction section.
2. In Figure 1, the final bullet point is difficult to read. It is suggested to revise that point.
3. The conclusion is quite short. The authors should expand this section to highlight the major points of the study (effect of ALA vs. ALA-PDT, limitations of the study, etc.)
Author Response
- The authors are studying the effect of ALA and PDT on exosomes, but the significance of exosomes is not presented in the introduction section.
We have added more information on the possible roles of exosomes in CD and ALA-PDT in the Introduction (para.-3 &-4, p-2) of our revised version.
- In Figure 1, the final bullet point is difficult to read. It is suggested to revise that point.
We have modified the Figure-1 (p-3) according to the comment in the revised version.
- The conclusion is quite short. The authors should expand this section to highlight the major points of the study (effect of ALA vs. ALA-PDT, limitations of the study, etc.).
We fully agree with the comment and have revised the Conclusion (p-18) in the revised version.
Reviewer 2 Report
This paper describes the effect of ALA-PDT on PBMC subpopulations in patients with active Crohn's disease (CD). The obtained results suggest that ALA-PDT may be a potential candidate for the treatment of CD and other immune-mediated diseases. The work is interesting and promising. However, there are some points for the authors' consideration:
1. The data analysis was not ideal. For example, there is no error bar in figure 1B. And all there is no significance analysis in all bar chart.
2. The sample screening process should be further clarified. why only 8 samples obtained the live cell count from 25 samples. Does this mean there is no live cells in most of the samples?
3. For figure 3 and 4, what does the control group mean? before treatment? or health controls?
4. Furthermore, from figure 3 and 5, it seems the ALA itself already induce broad changes in patients? So why the authors attribute the treatment effects to PDT?
5. Similarly, I can not tell from figure 4 and 6 that are all the changes in cytokines reaching the significance.
Author Response
- The data analysis was not ideal. For example, there is no error bar in figure 1B. And all there is no significance analysis in all bar chart.
We have added a full new paragraph (under 3.6 Section, p-17) in the revised version to explain how we sampled in the study and why no error bars in the Figure 2B. We did not do statistical p–value tests because we obtained too few (but valuable) clinical patient samples in this study, although the Figure 2C shows obviously significant differences in the survivals of monocytes in all samples after ALA-PDT.
- The sample screening process should be further clarified. why only 8 samples obtained the live cell count from 25 samples. Does this mean there is no live cells in most of the samples?
We have explained the criterion (>2 million viable cells) for selecting patient samples in the revised version (3.1 Section, p-15).
- For figure 3 and 4, what does the control group mean? before treatment? or health controls?
In the two figures the control group means the group treated with no ALA, nor light. We used CD patient PBMC samples, not healthy donor samples
- Furthermore, from figure 3 and 5, it seems the ALA itself already induce broad changes in patients? So why the authors attribute the treatment effects to PDT?
Since ALA alone had some effects on cytokines and exosomes, in order to see ‘pure’ effects of ALA-PDT on cytokines and exosomes we needed to subtract the effects of ALA alone.
- Similarly, I can not tell from figure 4 and 6 that are all the changes in cytokines reaching the significance.
Both ALA alone and ALA-PDT can change the levels of cytokines with a huge variability in different individual samples. It is thus difficult to evaluate if such changes are statistically significant or not in a study with a small number of samples.
Reviewer 3 Report
This work reported the effects of ALA-PDT on cytokines and exosomes of human healthy peripheral blood mononuclear cells (PBMCs) and investigated the ALA-PDT-mediated effects on PBMC subsets from the patients with Crohn’s Disease (CD). Firstly, the idea was not innovative. Meanwhile, what surprised me most was that there were two identical figures in the text (Figure 2C, i, ii). Figure 2A and 2C both had error bar but Figure 2B did not. Perhaps the authors forgot to modify the previous version, which made me doubt whether the manuscript had been finally proofread before it was submitted. Considering the idea and the low-level error in the version sent, I think this work is not matched to this journal.
Author Response
- Firstly, the idea was not innovative
This is the first time to study the effects of ALA-PDT on exosomes in PBMCs of clinical CD patients. The goal was to better understand possible mechanisms for our onging clinical trial of ALA-mediated photopheresis of CD patients, a first-in-human study.
- Meanwhile, what surprised me most was that there were two identical figures in the text (Figure 2C, i, ii).
We do appreciate the reviewer for his/her finding the careless mistake. We do apologize it!
- Figure 2A and 2C both had error bar, but Figure 2B did not.
We have added a full new paragraph (under 3.6 Section, p-17) in the revised version to explain how we sampled in the study and why no error bars in the Figure 2B.
Round 2
Reviewer 2 Report
The authors have addressed most of my comment. However, I still feel some points should be further explained before it can be published.
1. why the 2×106 live cells per sample was set as the threshold for this study. Normally 1*105 live cells even 1*104 live cells could give enough data for analysis. Please explain.
2. Representative Flow Cytometry images should be provided. Especially the process of how to gate the target cells should be given.
3. I still fell the patient number is too small to acheive the conclusion. Especially as explained by authors" statistical p–value tests could not be performed because they obtained too few (but valuable) clinical patient samples "
4. Last but not least, I did not see any figure under 3.6 Section, p-17.
Author Response
- why the 2×106 live cells per sample was set as the threshold for this study. Normally 1*105 live cells even 1*104 live cells could give enough data for analysis. Please explain.
We agree with the comment. We used 2 million viable cells per group and divided them equally into 4 samples: 5×105/sample for control, ALA alone, light alone and ALA plus light. This info has been added to the Section 3.5 of the revised version.
- Representative Flow Cytometry images should be provided. Especially the process of how to gate the target cells should be given.
We have followed the comment and added the gating strategy of flow cytometry in a supplementary file of the revised version.
- I still fell the patient number is too small to acheive the conclusion. Especially as explained by authors" statistical p–value tests could not be performed because they obtained too few (but valuable) clinical patient samples "
We have removed ‘significantly’ and changed the sentence to ‘ALA-PDT clearly killed monocytes’ in the Abstract and Section 2-2. In addition, we have mentioned the small number of samples in Conclusion.
- Last but not least, I did not see any figure under 3.6 Section, p-17.
In the previous revised version, we added a paragraph under Section 3.6 to explain how we have sampled (in Fig. 2) for measuring lymphocytes, B-cells and monocytes.
Reviewer 3 Report
This work reported the effects of ALA-PDT on cytokines and exosomes of human healthy peripheral blood mononuclear cells (PBMCs) and investigated the ALA-PDT-mediated effects on PBMC subsets from the patients with Crohn’s Disease (CD). I advise accept this article after minor revision (corrections to minor methodological errors and text editing)
Author Response
- I advise accept this article after minor revision (corrections to minor methodological errors and text editing)
We appreciate the comment and have tried our best to correct/revise the whole manuscript with respect to methodology including the gating strategy of flow cytometry as a supplementary file and also to text editing.
Round 3
Reviewer 2 Report
The authors have addressed my comments.